# Omphalocele and Cardiac Abnormalities—The Importance of the Association

**DOI:** 10.3390/diagnostics13081413

**Published:** 2023-04-14

**Authors:** Elena Țarcă, Dina Al Namat, Alina Costina Luca, Vasile Valeriu Lupu, Razan Al Namat, Ancuța Lupu, Laura Bălănescu, Jana Bernic, Lăcrămioara Ionela Butnariu, Mihaela Moscalu, Marius Valeriu Hînganu

**Affiliations:** 1Department of Surgery II-Pediatric Surgery, “Grigore T. Popa” University of Medicine and Pharmacy, 700115 Iassy, Romania; 2“Saint Mary” Emergency Children Hospital, 700309 Iassy, Romania; 3Department of Mother and Child Medicine–Pediatrics, “Grigore T. Popa” University of Medicine and Pharmacy, 700115 Iassy, Romania; 4Faculty of Medicine, “Grigore T. Popa” University of Medicine and Pharmacy, 700115 Iassy, Romania; 5Department of Pediatric Surgery and Anaesthesia and Intensive Care, “Carol Davila” University of Medicine and Pharmacy, 020021 Bucharest, Romania; 6Discipline of Pediatric Surgery, “Nicolae Testemițanu” State University of Medicine and Pharmacy, 2025 Chisinau, Moldova; 7Department of Medical Genetics, Faculty of Medicine, “Grigore T. Popa” University of Medicine and Pharmacy, 700115 Iassy, Romania; 8Department of Preventive Medicine and Interdisciplinarity, “Grigore T. Popa” University of Medicine and Pharmacy, 700115 Iassy, Romania

**Keywords:** omphalocele, cardiac abnormalities, cardiac malformation

## Abstract

Omphalocele is the most common ventral abdominal wall defect. Omphalocele is associated with other significant anomalies in up to 80% of cases, among which the cardiac ones are the most frequent. The aim of our paper is to highlight, through a review of the literature, the importance and frequency of association between the two malformations and what impact this association has on the management and evolution of patients with these pathologies. We reviewed the titles, the available abstracts, and the full texts of 244 papers from the last 23 years, from three medical databases, to extract data for our review. Due to the frequent association of the two malformations and the unfavorable effect of the major cardiac anomaly on the prognosis of the newborn, the electrocardiogram and echocardiography must be included in the first postnatal investigations. The timing of surgery for abdominal wall defect closure is mostly dictated by the cardiac defect severity, and usually the cardiac defect takes priority. After the cardiac defect is medically stabilized or surgically repaired, the omphalocele reduction and closure of the abdominal defect are performed in a more controlled setting, with improved outcomes. Compared to omphalocele patients without cardiac defects, children with this association are more likely to experience prolonged hospitalizations, neurologic, and cognitive impairments. Major cardiac abnormalities such as structural defects that require surgical treatment or result in developmental delay will significantly increase the death rate of patients with omphalocele. In conclusion, the prenatal diagnosis of omphalocele and early detection of other associated structural or chromosomal anomalies are of overwhelming importance, contributing to the establishment of antenatal and postnatal prognosis.

## 1. Introduction

Congenital anomalies are birth defects that lead to significant health problems and adverse outcomes in neonates and infants, often with problems that can persist into adulthood. The association of several birth defects can be even more serious as it potentially increases morbidity, infant mortality, and it also increases medical costs and prolongs hospitalization [1]. In general, 3–5% of newborns may have a congenital malformation, depending on the degree of socio-economic development of the respective country, the rate of antenatal diagnosis, and the rate of therapeutic abortion according to the specific laws [2,3,4,5]. Omphalocele is the most common ventral abdominal wall defect and is usually associated with other significant anomalies in up to 80% of cases, among which cardiac ones are the most frequent [6,7].

Since not all countries have a national registry for tracking congenital anomalies, the incidence and prevalence of omphalocele and cardiac malformations are not exactly known, much less their association. It seems that cardiac defects are some of the abnormalities more commonly associated with an omphalocele, between 7 and 50%, and have the most substantial impact on the survival and outcomes of the neonate [2]. When the omphalocele is associated with other congenital malformations such as diaphragmatic hernia, cardiac abnormalities occur with a frequency of up to 54%, especially if the omphalocele is large. Cardiac malformations appear to occur in 11% of cases of omphalocele less than 5 cm in diameter, and in up to 46% of cases of larger diameter omphalocele [8,9]. Omphalocele is a malformation that can be easily resolved surgically when the dimensions of the abdominal wall defect are small, but in the case of large defects, with externalized liver, and especially when associated with cardiac malformations, the morbidity and mortality rates may increase considerably.

The aim of our paper is to highlight, through a review of the literature, the importance of and frequency of association between the two malformations and what impact this association has on the management and evolution of patients with these pathologies. 

## 2. Materials and Methods

### Electronic Databases and Search Strategy

This review was performed according to the Preferred Reporting Items for Systematic Reviews and Metanalysis Statement (http://www.prisma-statement.org/ (accessed on 15 December 2022)). Four researchers performed independent data extraction and quality assessment. We used the PubMed, Web of Science, and EMBASE engines for our study, and searched for the association of omphalocele with cardiac abnormalities/malformation in published articles. The initial inclusion criteria were original studies written in English, systematic reviews, randomized controlled trials, observational studies, series of cases studies, and case reports, from the earliest time possible until December 2022.

## 3. Results

A total of 6608 results were obtained from PubMed, and only 1265 with abstract and free full text available. Then, we used the Medical Subject Headings (MeSH) terms “cardiac abnormalities” AND “cardiac malformation” to select the appropriate articles for our review. In the mentioned order, 102 and 120 articles were found (90 articles were found to be eligible for our review in both searches, based on title and abstract assessment).

From Web of Science, 2089 articles were found discussing the omphalocele topic; overall, 115 results were obtained after searching for “omphalocele” AND “cardiac abnormalities” OR “omphalocele” AND “cardiac malformation”, from the last 23 years. We applied Web of Science category filters (Cardiac and Cardiovascular System, Pediatrics, Obstetrics and Gynecology, Genetics Heredity and Surgery) and analyzed the resulting 87 articles.

A total of 4261 articles were obtained from EMBASE for the omphalocele search; after using the association “cardiac abnormalities” OR “cardiac malformation”, 108 and 362 articles were identified. Of these, only 70 and 264 were human studies and reviews (filters applied). After removing the duplicates and sorting for relevance, 67 articles were selected from EMBASE for our review (Figure 1).

### Literature Review

During the second step, we reviewed the titles, the abstracts, and the full texts of 244 papers from the last 23 years to extract pertinent data for our review. Eighty-eight relevant studies with their citations were included in our reference list.

The frequent association of omphalocele with other congenital malformations and, respectively, with cardiac malformations is explained by the analysis of embryological development. An adverse event (toxic or teratological factors, genetic anomalies) that occurs in weeks 8–10 of gestation, which are the weeks of maximum development of both the primitive intestine, the anterior abdominal wall, and the heart in parallel, will affect all these organs and will result in associated anomalies [10]. If the teratogenic defect involves the cephalic fold, the result will be a supraumbilical coelosomia (epigastric omphalocele associated with lower sternal anomalies, cardiac malformations, absence of pericardium, and diaphragmatic defect, all known as Cantrell’s pentalogy). Therefore, in the presence of a low sternal cleft, even a small one, the malformation is a superior coelosomy. If the caudal fold is the affected one, a lower coelosomia will form. The absence of closure of the abdominal wall in the hypogastric region (hypogastric omphalocele, vesico-intestinal fissure, bladder exstrophy, cloacal exstrophy) is also associated with a developmental defect of the pelvic girdle [10]. Thus, in older theories, omphalocele occurs as a result of developmental arrest of the body cavity between the 8th and 12th weeks of gestation; consequently, the herniated midgut is unable to return back [11]. A newer theory suggests that defective cell–cell signaling at the critical transition line between the amnion and the ventral body wall will lead to the failure of migration and fusion of the body folds at the umbilical ring, as well as to the failure of formation and migration of the abdominal bands and ventral musculature [12]. Today, the accepted theory for the development of an omphalocele is the embryonic dysplasia theory combined with a malfunction of the ectodermal placodes [13].

Although no specific chromosomal abnormality has been demonstrated to be involved in the etiology of the omphalocele, there are studies that estimate that about 50% of fetuses with omphalocele have associated abnormal karyotypes, and trisomy 18 is the most common [8,14]. In homozygous knockout mice, Ma and Adelstein, when studying the expression of non-muscle myosin (variant IIB), it was found that a point mutation (R709C) of the gene *Myh10* is associated with cardiac abnormalities and midline defects (cleft palate, ectopia cordis, omphalocele, diaphragmatic hernia) because of a dominant negative effect [15]. Anyway, more research is needed to demonstrate the role of the mutations in non-muscle myosin variant IIB and related proteins in human patients with the diagnosis of pentalogy of Cantrell. 

The most frequent cardiac abnormalities associated with omphalocele are septal defects, tetralogy of Fallot, tricuspid atresia, or ectopia cordis [16]. Prenatal ultrasound diagnosis of an omphalocele without liver is certain after 12 weeks of gestational age, but extra-abdominal liver tissue could be observed transvaginally at 9–10 weeks of amenorrhea, while an elevated maternal serum alpha-fetoprotein level may also be detected [17].

## 4. Epidemiology

The prevalence of omphalocele is 2.5–4.0 per 10,000 births, and into the gestation period, 1.0–2.6 per 1000 fetuses, with a female–male ratio between 1:1.17 and 1:1.9 [13,18]. For the United States of America, the national birth prevalence estimates per 10,000 live births ranged from 0.62 for interrupted aortic arch to 19.93 for the 12 critical congenital heart defects combined. In addition, the prevalence for atrioventricular septal defect (AVSD), tetralogy of Fallot, and trisomy 18 increased in the 2010–2014 period compared to the previous periods (1999–2001 and 2004–2006) [1]. For omphalocele, the estimated prevalence in this study was 2.45 (1/4175 birth), increasing in comparison with previous years. In any case, when aborted pregnancies or stillbirths are also taken into consideration, the number of these anomalies increases significantly. 

An omphalocele patient can present with a cardiac malformation in 10–50% of the cases, and it seems that the association rate increases in children with large omphaloceles (>4 cm) [2,9]. Interestingly, it is more common to find chromosomal abnormalities in cases of small defects rather than when a giant omphalocele is present [19].

During fetal ultrasounds performed in the second trimester, approximately 35% of fetuses with omphalocele have an associated congenital heart disease, and 59% of them have an abnormal cardiac axis [20]. In utero death may occur if, in addition to omphalocele and cardiac malformation, the fetus also has chromosomal abnormalities or other severe congenital malformations. Depending on the legislation in each country, therapeutic abortion may be recommended. The rate of elective termination of pregnancy may reach 50% in some countries [14]. In this way, the association rate of the two malformations may change postnatally, and even if the karyotype is normal, the incidence of cardiac defects in patients with omphalocele varies between 19 and 41.6% [6,14,21]. A study performed in the Netherlands identified an 83% rate of multiple associated anomalies, and, of these, 57% had an abnormal karyotype (most commonly trisomy 18). Of 141 fetuses with omphalocele, 61% underwent termination of pregnancy and another 18% underwent spontaneous miscarriage; only 24 (17%) of the detected fetuses were born, and 12 (50%) had an isolated omphalocele [3].

## 5. Omphalocele Patients with Cardiac Abnormalities

It is important to note that even in the setting of a normal karyotype, at least 50% of fetuses with omphalocele may have other abnormalities, and the cardiac ones are the most frequent. This is why the American Heart Association recommends performing a fetal echocardiogram in all fetuses when extra cardiac abnormalities are present [22]. Septal defects are the most frequent cardiac abnormalities associated with omphalocele, followed by tetralogy of Fallot, ectopia cordis, tricuspid atresia, hypoplastic left heart syndrome, and others [6]. A recent study demonstrated a 42.4% rate of cardiac abnormalities associated with omphalocele, but only 20.8% of all omphalocele patients had major cardiac malformations (ventricular septal defect, severe pulmonary valvular stenosis, tetralogy of Fallot, transposition of the great arteries, hypoplastic left heart syndrome, atrioventricular canal) [18].

Major cardiac abnormalities such as structural defects that require surgical treatment or result in developmental delay will significantly increase the mortality of patients with omphalocele [8,23]. Marshall et al. established a 2.4 times higher mortality rate in the first year of life for infants with associated omphalocele and major cardiac defects compared to infants with isolated omphalocele [2]. If the analysis includes all cardiac abnormalities, they do not seem to significantly affect the survival rate of children with omphalocele, but when omphalocele is associated with more than one abnormality (including a minor cardiac defect), the mortality rate increases 6.28 times [18]. In a study from 2018, Rees et al. demonstrated that the presence of major cardiac anomalies does not affect the surgical outcome of neonates with exomphalos major; however, this study has the limitation of a small number of patients (only 22 patients) [24]. Another study on 35 giant omphalocele demonstrated increased odds of unfavorable outcomes in the presence of major cardiac anomalies [23]. In a group of 123 patients, 59 of whom had giant omphalocele, Elhedai et al. demonstrated that deaths in the giant omphalocele group associated with congenital heart disease had mainly right cardiac anomalies, and all these neonates required mechanical ventilation for pulmonary hypoplasia before cardiac surgery. In contrast, those who survived did not require mechanical ventilation prior to cardiac intervention. The 5-year survival was 20% for the giant omphalocele associated with cardiac abnormalities versus 90% for those with minor or no heart disease (*p* < 0.0001) [25].

### 5.1. Atrial Septal Defects (ASD)

Atrial septal defects are found with an incidence of 1.3 per 1000 live births and are usually asymptomatic, being detected on auscultation of the heart followed by echocardiography. If the defect is small, medication is recommended and the omphalocele surgery may be instituted. Large atrial septal defects can become symptomatic, cause heart failure, or cause growth retardation. Then, closure is needed, and ASD can be treated with catheter-based interventions or open cardiac surgery; the repair carries a minimal risk, with mortality risk being much less than 1% [26]. 

### 5.2. Ventricular Septal Defect (VSD) 

Ventricular septal defects are found with an incidence of 4.2 per 1000 live births, and, if small, they may be asymptomatic; in this case, VSDs are managed conservatively [6]. Large VSDs usually manifest within 3–4 weeks of life, causing congestive heart failure [26]. If the VSD is large and needs surgical intervention, then it is better for the patient to apply conservative treatment for omphalocele, meaning a “paint and wait” technique [8]. The omphalocele will be transformed into a ventral hernia, and after the treatment of VSD and stabilization of the cardio-vascular system, the ventral hernia may be surgically repaired (usually after the age of one). 

### 5.3. Hypoplastic Left Heart Syndrome

Hypoplastic left heart syndrome is a major congenital malformation (severe form of cyanotic congenital heart disease) and has been reported to occur in approximately 0.016 to 0.036% of all live births. It may be associated with underdevelopment of the aorta and aortic arch, as well as mitral stenosis or aortic atresia; The Norwood procedure is recommended as the treatment in the neonatal period. As the arterial duct closes, the systemic perfusion decreases, resulting in hypoxemia, acidosis, and shock [27]. Until correction, medication is necessary (prostaglandins) to keep the ductus arteriosus open; additionally, the omphalocele closure will be delayed [28]. If primary cardiac transplantation is needed and the omphalocele is large, the prognosis is very poor. 

### 5.4. Tricuspid Atresia (TA)

Tricuspid atresia is a frequent cardiac anomaly associated with omphalocele; although it has a small incidence in the general population, it accounts for 0.47% of all cardiac defects antenatally detected, and 0.1 per 1000 live births [29]. The anomaly is a severe one, with outcomes for TA being similar to those for hypoplastic left heart syndrome. TA is defined as an absent/imperforate right atrioventricular connection, frequently associated with pulmonary stenosis or atresia, or hypoplasia of the aortic arch. The prenatal diagnosis of TA and its associated intra- and extra-cardiac anomalies is now possible with a high degree of accuracy, and termination of pregnancy may be recommended in the presence of chromosomal abnormalities or giant omphalocele [29,30]. The neonate will receive prostaglandins to keep the ductus arteriosus open, and then staged surgical interventions are necessary. The survival rate after birth is up to 89% in the neonatal period, and 60% after 20 years [29].

### 5.5. Ectopia Cordis (EC)

Ectopia cordis is the complete or partial displacement of the heart outside the thoracic cavity, associated with defects of the pericardium, diaphragm, sternum, and cardiac malformations. In addition, it may be associated with chromosomal abnormalities (trisomy 18, Turner syndrome, 46,XX, and 17q+) and other congenital malformations such as omphalocele; the occurrence prevalence is between 5.5 and 7.9/million live births [31]. Depending on the location of the heart, it is classified into different types (cervical, thoracic, and abdominal); the thoracoabdominal type is regarded as a distinct syndrome known as the pentalogy of Cantrell. Newborns with these complex malformations require intensive care and immediate surgical management to cover the exposed heart and viscera. It is obvious that the cardiac anomaly has priority over the surgical treatment of the omphalocele, and after covering the pericardial defect and closing the chest wall defect, the management of the case will be established according to the cardiac structural anomalies. The prognosis is usually poor, but patients may survive to adulthood if they have an incomplete EC, fewer intracardiac defects except for ASD, and absence of an omphalocele [32].

### 5.6. Associated Abnormalities of Systemic Veins

Associated abnormalities of systemic veins may also be detected, even antenatally. In a retrospective study carried out over a period of 16 years, 34 out of 9627 fetuses examined by ultrasound for a possible cardiovascular anomaly had omphalocele; overall, 7 of the 34 presented with an interrupted inferior vena cava with azygos continuation to a right-sided superior vena cava, and only one of them had cardiac structural abnormalities. The omphalocele was large and contained liver in all seven cases [33].

### 5.7. Pulmonary Hypertension and Right Ventricular Dysfunction

Pulmonary hypertension and right ventricular dysfunction have been described in the neonatal omphalocele population, and are worse in neonates with a giant omphalocele. This is due to the abnormalities in the pulmonary parenchyma and vasculature that lead to pulmonary hypoplasia [34]. Right ventricular dysfunction at initial echocardiography was significantly associated with mortality [35]. This is why early and serial assessment of pulmonary hypertension by echocardiography must be performed as a routine procedure, especially in the giant omphalocele population who require a staged surgical repair procedure [36].

## 6. Chromosomal Abnormalities and Syndromes

A possible explanation for the increase in the prevalence of omphalocele and major cardiac defects is the increased prevalence of trisomy 13, 18, and Down syndrome [1]. Trisomy 18 is the most frequent chromosomal anomaly, where as many as 80–90% patients have an omphalocele [8]. Since the frequency of occurrence of omphalocele in fetuses with trisomy 13 and 18 exceeds 20%, while in those without chromosomal abnormalities it is 0.05%, the calculated risk of trisomy in fetuses with omphalocele is 340 times higher than in those without omphalocele [37]. The risks of trisomy 13 and 18 increase with maternal age, and since these chromosomal abnormalities are associated with an increased rate of intrauterine death, their prevalence decreases with gestational age. Consequently, both the prevalence of omphalocele and chromosomal abnormalities will increase with maternal age, and decrease with increasing gestational age. There are studies that have analyzed fetuses with Turner syndrome and small omphalocele; the affected fetuses are at a high risk of miscarriage and intrauterine fetal death, especially with the karyotype 45,X, and if fetal hydrops is present. In a study on thirty-five fetuses with Turner syndrome and omphalocele, the detected cardiac anomalies were VSD/AVSD (5/35 cases), AVSD with coarctation of the aorta (1/35), hypoplastic left heart (3/35), coarctation of the aorta in one case, and pulmonary insufficiency in another case (at 25 weeks GA) [38]. In these cases with associated cardiac abnormalities, the pregnancy outcome was poor. 

Performing an antenatal karyotype should be part of the investigation protocol in the case of an early diagnosis of omphalocele, because studies performed on fetuses diagnosed antenatally detect karyotype changes in 23–54% of cases, with these percentages varying depending on the time of detection. Thus, the prevalence of chromosomal abnormalities associated with omphalocele is approximately 30% for cases detected in the first trimester of pregnancy, and only 10% at birth, the explanation being that most of the fetuses with chromosomal abnormalities associated or not with an omphalocele will be spontaneously aborted until the end of the pregnancy [5]. A study carried out in 2008 detects a 29% frequency of chromosomal abnormalities in fetuses with omphalocele [39]. Central omphaloceles are more strongly associated with abnormal karyotype (69%) than epigastric omphaloceles (12.5%) are [14].

The syndromes associating omphalocele and cardiac defects are PAGOD syndrome (pulmonary hypoplasia, agonadism, omphalocele, dextrocardia, and congenital diaphragmatic hernia), Beckwith–Wiedemann syndrome, OEIS (omphalocele, exstrophy, imperforate anus, spinal defects), left atrial isomerism, pentalogy of Cantrell, and others.

### 6.1. PAGOD Syndrome

PAGOD syndrome is an extremely rare congenital malformation complex which includes omphalocele, pulmonary artery and lung hypoplasia, diaphragmatic defects, sex reversal, or ambiguous genitalia and dextrocardia; other types of cardiac abnormalities which may be included are persistent ductus arteriosus, atrial or ventricular septal defects, mitral or aortic atresia, Wolff–Parkinson–White (WPW) Syndrome, hypoplastic left heart, and scimitar syndrome [40,41]. According to Rossetti et al., although PAGOD syndrome is considered to be an autosomal or x-linked disorder, individuals with this diagnosis and a positive family history may possibly have a myelin regulatory factor gene (*MYRF*) deficiency [41]. They also draw attention to the importance of monitoring patients with *MYRF* deficiency surviving the neonatal period for developmental delay and intellectual disability. There are only few cases reported in the literature to date; from 1991, when it was described by Kennerknecht et al. and until 2014, only 11 cases were described [42,43]. One patient with PAGOD syndrome suffered coarctation repair but died at home with WPW syndrome [44]; another patient surviving a Norwood palliation for hypoplastic left heart and bidirectional cavopulmonary anastomosis, as a second stage of operation, was waiting for a Fontan procedure in Kim et al.’s report [40]. Another patient died at the age of 39, days after surgical repair of diaphragmatic eventration and aortic arch hypoplasia, and despite intensive support including extracorporeal membrane oxygenation veno-arterial (ECMO-VA) [43].

### 6.2. Beckwith–Wiedemann Syndrome (BWS)

Beckwith–Wiedemann syndrome is a disorder characterized by omphalocele, cardiomyopathy, neonatal hypoglycemia, macrosomia, macroglossia, hemihypertrophy, renal abnormalities, embryonal tumors (such as Wilms tumor or neuroblastoma), and other abnormalities. Between 30% and 79% of patients with Beckwith–Wiedemann syndrome will have an omphalocele, and BWS occurs in approximately 15% of live born omphalocele cases, especially in association with small omphaloceles [45,46]. The incidence of the syndrome is 1/12,000 live newborns; it can have a familial incidence with autosomal dominant transmission, with incomplete penetrance and variable expressivity. There is an emerging association of assisted reproductive techniques (ART) with imprinting disorders, and it seems that ART entails a 10-fold increased risk of BWS, resulting in the need for awareness of ART-associated health risks [47]. In order to be able to establish the prenatal diagnosis, it is necessary to fulfill two major criteria, established by ultrasound (macroglossia, macrosomia, anterior abdominal wall defect), or one major criterion together with two minor criteria (nephromegaly/dysgenesis/dysplasia, adrenomegaly, aneuploidy/locus abnormal, polyhydramnios) [37]. Antenatal diagnosis is of particular importance for family counselling, pregnancy and birth planning, and post-natal management. The reported mortality of 15–20% is due to complications of prematurity, cardiomyopathy, malignant tumors, or hypoglycemia [48]. Glucose monitoring should be established in all neonates with features of Beckwith–Weidemann syndrome to reduce the risk of central nervous system complications.

### 6.3. Pentalogy of Cantrell (PC)

Pentalogy of Cantrell (PC), also named thoracoabdominal ectopia cordis, is a sporadic anomaly, an association of congenital midline birth defects, including heart, diaphragmatic pericardium, anterior diaphragm, lower sternum, and supraumbilical abdominal wall (omphalocele). The incidence is between 1 in 65,000 and 1 in 200,000 live births, with a slight male predominance of 1.35:1 [49]. Cardiac malformations are usually severe, with ventricular or atrial septal defect, tetralogy of Fallot, or pulmonary stenosis being the most common. Genetic abnormalities such as trisomy 13, 18, 21, and Turner syndrome may be present [50]. PC may present in different forms, complete or incomplete, and other organs may be involved: the head and neck, the kidneys, the limbs. In 1958, James R. Cantrell presented five cases of his own (from twenty-one totals at that time, dating back to the early 1700’s) and defined this pentalogy consisting of a specific constellation of congenital abnormalities [51]. He published the article “A Syndrome of Congenital Defects Involving the Abdominal Wall, Sternum, Diaphragm, Pericardium, and Heart”, and described the abnormalities involved and postulated their embryologic developments and possible treatment [51]. In 1972, Toyama proposed a three-group classification: certain (all five defects are present), probable (four defects present, including an intra-cardiac abnormality and ventral wall defect), or incomplete pentalogy (often lacking the intra-cardiac defect, or one or more of the remaining defects) [52]. The cause of the emergence of PC is a defect in the development of the septum transversum, somatic and splanchnic mesoderm between 14 and 18 days of embryonic life, resulting in a defect of the anterior diaphragm, inferior pericardium, and cardiac structures. In addition, the primordial sternum fails to migrate and fuse fully, and there is an improper attachment of the abdominal musculature leading to omphalocele appearance [51]. If an epigastric omphaloceles is identified at a fetal ultrasound screening, the obstetrician will look for the presence of a sternal or pericardial defect, which are components of the pentalogy of Cantrell or its variants, and will further recommend 3D ultrasound, fetal echocardiography, or fetal MRI to provide a more detailed anatomic survey. The outcomes of these cases are determined more by the associated cardiac defect, as the ventral diaphragmatic hernia does not contribute much to the pulmonary morbidity; anyway, the survival rate remains as low as 37–61%, with few patients surviving through their early days of life, especially in the complete form [19].

### 6.4. Left Atrial Isomerism (LAI)

Left atrial isomerism is a heterotaxy syndrome that may associate omphalocele with polysplenia, congenital heart defects (ventricular noncompaction, ventricular hypertrophy, bradycardia, cardiomegaly, and heart block), bilateral bilobed lungs, and situs abnormalities of abdominal organs [53]. Splenic torsion is an unusual but dangerous condition that may occur in these patients, and physicians should be vigilant of the differential diagnosis of the acute abdomen [54]. 

### 6.5. OEIS Syndrome

OEIS syndrome is an inferior coelosomia associating a hypogastric omphaloceles with anomalies of the bladder (exstrophy), spine, anus, heart, etc. In 1978, Carey described for the first time the combination of omphalocele and bladder exstrophy, spinal defect, and imperforate anus, naming this association the OEIS complex [55]. The estimated incidence is 1/200,000–400,000 newborns, although some authors suggest a higher incidence, but the cases are underdiagnosed due to the fact that they do not meet all the criteria for a positive diagnosis [7]. The OEIS complex arises from a localized defect in the early development of the mesoderm that will later contribute to infraumbilical mesenchyme, cloacal septum, and caudal vertebrae. Most cases occur spontaneously, but several cases of intra-familial recurrence have been reported, as well as occurrence in mono- and dizygotic twins, suggesting a genetic contribution to the pathogenesis of OEIS syndrome. When detecting a fetus with an OEIS complex, the important organs (brain, heart, liver, kidney, limbs, umbilical artery and vein) should be carefully examined to describe the presence of associated malformations. Antenatal diagnosis (fetal ultrasound and elevated maternal serum alpha fetoprotein level) should lead to a therapeutic termination of pregnancy for fetal reasons [56]. If the diagnosis is uncertain, a fetal MRI may be helpful to provide accurate information on the structural defects of the cloaca, the spinal and cardiac defect, the abdominal wall defect, and the pathologic status of the bowels [57]. Differential diagnoses of OEIS syndrome include other abdominal wall defects such as schisis defects, exstrophy of the cloaca, and limb–body wall complex disorders [58].

### 6.6. Limb Body Wall Complex (LBWC)

Limb body wall complex (LBWC) developmental abnormalities during the embryonic disc period can lead to the appearance of the “limb body wall” complex, which associates an anterior abdominal or thoracic wall defect with severe scoliosis and other anomalies such as encephalocele, labio-maxillo-palatine clefts, cardiac or uro-genital malformations, limb anomalies (varus equine foot, syndactyly, amelia, arthrogryposis), amniotic bands, and placenta anomalies in intrauterine life. Cardiovascular anomalies include primitive ventricle, common atrium, atrial and ventricular septal defects, truncus arteriosus, hypoplastic right ventricle, and ectopia cordis. LBWC was described for the first time by Van Allen et al. in 1987 [59]. The most accepted theory is early embryonal dysplasia, but due to the 40% presence of amniotic bands, some authors argue that this complex is only a severe form of amniotic disease [60]. Ultrasound detection in the first trimester of pregnancy of the association of an omphalocele with scoliosis will raise the suspicion of LBWC, and will prompt additional investigations such as measurement of the maternal serum alpha fetoprotein level. Karyotyping is usually normal [61]. Early diagnosis followed by medical termination is the preferred treatment for this anomaly because it is considered that the malformation is incompatible with life. The anomaly is found equally in both sexes, and the incidence of this association is 0.21–0.31/10,000 live newborns [62].

### 6.7. The VACTERL Malformation Association

The VACTERL malformation association, in addition to the classic vertebral, anorectal, cardiac, tracheo-esophageal, renal, and limb anomalies, may also include omphalocele. All these anomalies can be diagnosed antenatally by ultrasound of the pregnant uterus or the analysis of certain markers in the amniotic fluid or maternal serum. Among all the malformations present in this case, the identification and ligation of the tracheo-esophageal fistula have priority as a surgical treatment. In the same surgical stage, esophageal anastomosis can be performed if the gap between the two esophageal ends is not too long. If it is a case of long-gap esophageal atresia, then a gastrostomy will be performed, and, at the same time, the anterior abdominal wall defect will be closed. In most cases, the cardiac anomaly is a minor one, usually an atrial or ventricular septal defect, but cases of right aortic arch or other abnormalities of the aortic arch can also be found. In these cases, identifying the anomaly antenatally or immediately postnatally is important for the pediatric surgeon because the thoracotomy or toracoscopy may be performed on the left side, or the esophageal anastomosis may be curtailed [63,64].

### 6.8. The ADAM Sequence (Amniotic Deformitis, Adhesions, Mutilations)

The ADAM sequence from the disease of the amniotic bands (Ombredanne disease) can associate defects of the anterior abdominal wall or of the thoracic cage with reduction anomalies of the limbs and sometimes cardiac, cranio-facial, vertebral, and other internal organ defects [65].

### 6.9. Otopalatodigital Spectrum Disorders

Otopalatodigital spectrum disorders include Otopalatodigital syndromes types 1 and type 2, Melnick–Needles syndrome, and frontometaphyseal dysplasia. These syndromes constitute a group of dominant X-linked osteochondrodysplasias and are associated with variable skeletal dysplasia in male children (more frequently than females), brain malformations, palatoschisis, cardiac malformations, omphalocele, and obstructive uropathy [66].

In Table 1, we have summarized the incidence of the most common cardiac abnormalities associated with omphalocele, as well as the presence of this association in some syndromes.

## 7. Management

The specific management of omphalocele or cardiac malformations is not the scope of this article. We highlighted only the particularities of the treatment of newborns that associate the two malformations. 

In the case of small and medium omphalocele, natural birth has no contraindications. The timing and mode of delivery should be based on standard obstetric indications. In the case of a giant omphalocele with a herniated liver, a ruptured sac, and especially when a severe cardiac malformation or an ectopia cordis is associated, cesarean delivery is recommended to avoid herniated viscera damage and prolonged cardiac compression [69,70]. Barring any fetal complications, preterm delivery is not recommended, the consensus being that the pregnancies should be allowed to proceed to term, but the delivery will be planned to take place in a tertiary hospital [19].

If it is large, the omphalocele is a serious congenital malformation, but it does not put the newborn’s life in immediate danger. When it is associated with other abnormalities, especially cardiac ones, biological and imaging investigations are urgent. Prenatal diagnosis of omphalocele can be established by ultrasound as early as 11 to 14 weeks gestation in tertiary level referral centers. Once the diagnosis of omphalocele is established, other congenital anomalies will be looked for and the cardiac axis will be measured, this being abnormal in 59% of fetuses with omphalocele; cardiac axis abnormalities are a sign of the presence of a structural cardiac abnormality [20]. The sensitivity for fetal echocardiograms to detect complex congenital heart defects is greater than 80% [71]. Anyway, if a complex cardiac malformation is found on fetal ultrasound, and especially if the ventricular septum is involved, fetal MRI may be recommended for a better evaluation and for family counseling [6,22,72]. In the presence of the association of several congenital anomalies, the mother will be directed to a tertiary center specialized in maternal–fetal medicine, both for counseling, follow-up, pre-, and postnatal treatment. From the point of view of prenatal treatment, few interventions can be performed safely, although Engels reported a case of PC in which he performed a pericardio-amniotic shunt for large pericardial effusion, with good results [73]. The survival rate in patients with omphalocele and cardiac anomalies detected antenatally is 23% compared to an 80% rate in live birth cases detected postnatally [14].

Regardless of whether or not the cardiac anomaly was visualized by ultrasound or by fetal MRI, the usual rule is to perform a transthoracic echocardiogram prior to any planned surgical intervention. This is to exclude or confirm any structural anomaly of the heart, and also to detect and monitor any significant pulmonary hypertension (standard criteria include elevated right ventricular systolic pressures and septal flattening). Pulmonary hypertension is usually associated with other congenital anomalies, longer durations of mechanical ventilation, and the need for high-frequency oscillatory ventilation and/or tracheostomy [74]. However, a recent study demonstrated that critical congenital heart defects were not missed on any patient with an omphalocele who had a normal fetal echocardiogram, so a routine postnatal transthoracic echocardiogram may not be needed in those neonates with a normal clinical workup [22]. If a murmur is heard during auscultation, if an abnormal EKG or abnormal chest X-ray are detected on postnatal evaluation, or desaturation based on pulse oximetry is noted, and if signs and symptoms of congestive heart failure appear, then further testing and echocardiography should be performed to guide patient management [75]. A plain chest X-ray may reveal dextrocardia or displacement in cases of ectopia cordis, or the presence of a congenital diaphragmatic hernia. Thoracoabdominal computed tomography (CT) or CT-angiography may be required to provide additional information regarding the intra-cardiac, diaphragmatic, pericardial, or intra-abdominal defects. 

In those neonates with omphalocele and a severe cardiac malformation requiring surgery or resulting in developmental delay, the probability of death is higher [6]. If the cardiac malformation requires open surgical intervention or catheter-based interventions, usually this must be performed before closing the abdominal wall defect, because reduction of the herniated viscera inside the peritoneal cavity may alter preload and afterload, leading to hemodynamic instability; additionally, if the liver is outside the abdomen, suprahepatic veins may be torsioned during the integration of the liver and cardiac, or liver insufficiency may appear. The need to perform several surgical interventions, parenteral nutrition, blood transfusion, mechanical ventilation, as well as prolonged hospitalization will favor the occurrence of nosocomial infections and will increase the morbidity rate [45,69,76,77]. In any case, the order of surgical interventions and case management will be established by a multidisciplinary team. When the cardiac malformation is not surgical, drug treatments such as spironolactone, prostaglandins, or other cardiac supports are recommended [26]. 

For children with BWS, management consists in the prevention and treatment of hypoglycemia, abdominal wall repair for omphalocele, cardiologic monitoring and treatment of cardiomyopathy, surgical or medical treatment of associated abnormalities and embryonic tumors, genetic testing, and counseling [48]. These children also present with macroglossia and are a high-risk group for obstructive sleep apnea, especially those younger than 6 months. Because continuous positive airway pressure at home can be challenging for some families, early tongue reduction may be an effective and safe option for these patients [78]. 

The management of neonates with OEIS complex is challenging and also not standardized; this is because of clinical instability secondary to comorbidities, including cardiac ones. Early and complex surgeries carry a high risk of death, and delaying first-stage exstrophy repair to allow physiologic optimization is safer. Repairs must be delayed secondary to cardiac conditions, neurosurgical interventions, and medical disease [79].

Neonates with PC require multi-disciplinary team management in a tertiary hospital and a staged approach is preferable when there are multiple malformations; if the neonate is unstable, early surgical intervention may be a risk factor for mortality, and management should be conservative: prophylactic antibiotics, and daily dressing changes to allow epithelialization of the omphalocele sac [8,49]. After stabilization, PC treatment is largely surgical, consisting in the separation of the abdominal and pericardial compartment, repair of the diaphragm and the anterior abdominal wall defects, as well as the cardiac malformations and malposition [50]. Cardiac anomalies are more important and are corrected first to prevent cardiac trauma and so that cardiac function is not affected by the reconstruction of the thoraco-abdominal wall. The staged repair may significantly reduce postoperative respiratory insufficiency, ventilator dependency, and mortality, although in cases of incomplete PC, the repair of all defects in a single surgical intervention can be feasible and safe [80,81]. Both regarding the integration of the heart in the small thoracic cavity and the integration of the intestinal loops with closure of the anterior abdominal wall defect, these will be carried out with great care in order to avoid twisting the organs and creating an increased postoperative intrathoracic or intraabdominal pressure. Big defects can be closed primarily with autologous grafts, or may be covered with the help of a biocompatible or polytetrafluoroethylene (Gore-Tex) patch [69,82]. In cases of severe cardiac defects with failure to thrive or severe pulmonary stenosis, palliative procedures are performed, and the complete repair of the heart is postponed. Heart failure, arrhythmia and embolism, cardiac rupture, cardiac tamponade, and endocarditis are the main complications and causes of death [83]. Late mortality is due to complications of cardiac dysfunction, infections, respiratory insufficiency, and adhesive small bowel obstruction [69,82].

The survival rate for newborns with isolated omphalocele is between 50% and 95%, but it significantly decreases to 30–68% when other anomalies are present [8,18,80,84]. In high-income countries with well-established health care systems and maternal–fetal care, the high rates of prenatal diagnosis of omphalocele associated with other abnormalities contribute to the high rates of pregnancy termination [85,86,87]. In these countries, the survival rates of newborns with omphalocele are high, even for those who have some cardiac abnormalities associated (positive selection of cases, but also due to improvements in neonatal intensive care). In low-income and middle-income countries, the prenatal detection of cases is scarce and the overall survival rates are as low as 30–68%, depending on the association of other abnormalities [85,86,88].

Limitation of the study: there are many retrospective, prospective studies, case series, and case reports from the earliest time possible until present, and we do not claim to have analyzed all of the available literature. Due to the marked heterogeneity of the form of presentation and treatment of the omphalocele, as well as due to the multitude of possibilities of malformative associations, we unfortunately cannot propose a standardized therapeutic approach. Therefore, each individual case must be carefully investigated, and the pre- and postnatal management indications must be specific. In any case, large omphalocele associated with major cardiac abnormalities has a severe prognosis, and therapeutic abortion can be considered.

## 8. Conclusions

Due to the frequent association of the two malformations and the unfavorable effect of the cardiac anomaly on the prognosis of the newborn, the electrocardiogram and echocardiography must be included in the first postnatal investigations. The timing of surgery for abdominal wall defect closure is mostly dictated by cardiac defect severity, and usually the cardiac defect takes priority. After the cardiac defect is medically stabilized or surgically repaired, the omphalocele reduction and closure of the abdominal defect are performed in a more controlled setting, with improved outcome. Compared to omphalocele patients without cardiac defects, children with this association are more likely to experience prolonged hospitalizations, neurologic, and cognitive impairments. That is why the prenatal diagnosis of omphalocele and early detection of other associated structural or chromosomal anomalies are of overwhelming importance, contributing to the establishment of antenatal and postnatal prognosis.

## Figures and Tables

**Figure 1 diagnostics-13-01413-f001:**
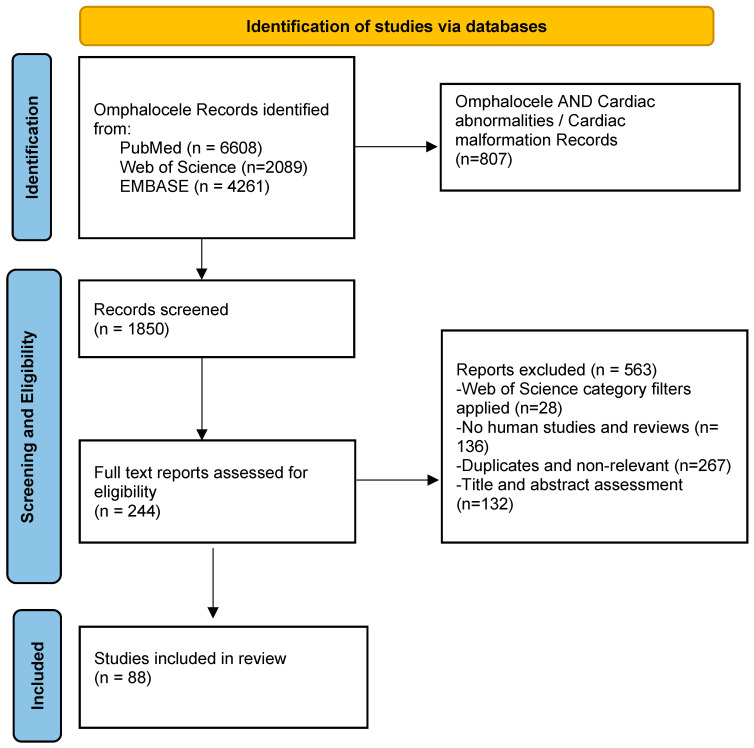
PRISMA flowchart.

**Table 1 diagnostics-13-01413-t001:** Cardiac defects associated with omphalocele and syndromes [6,38,67,68].

Cardiac Defect	Incidence	Mortality	Syndromes or Chromosomal Abnormalities
Atrial septal defects	1.3 per 1000 live births	Less than 1%	Trisomy 13, 18, and Down syndrome, Turner syndrome, PAGOD syndrome, VACTERL malformation association
Ventricular septal defect (VSD)	4.2 per 1000 live births	Surgically closed VSD displayed mortality rates of 1% and up to 3% if left untreated	Trisomy 13, 18, 21, Turner syndrome, PAGOD syndrome, Limb body wall complex, Pentalogy of Cantrell
Hypoplastic left heart syndrome	0.016 to 0.036% of all live births	About 20% to 60% of babies survive their first year of life	Turner syndrome, PAGOD syndrome
Tricuspid atresia	0.1 per 1000 live birth	The mortality rate after birth is up to 11% in the neonatal period and 40% after 20 years	VACTERL malformation association, Limb body wall complex
Ectopia cordis	5.5 to 7.9/million live births	Mortality exceeded 50% for infants <2500 g and <37 weeks of gestation.	Pentalogy of Cantrell, Limb body wall complex
Associated abnormalities of systemic veins	0.3% to 0.4% of the general population	Mortality up to 40%, depending on the type of abnormalities and associated defects	ADAM sequence, left atrial isomerism

VSD = Ventricular septal defect; PAGOD = pulmonary hypoplasia, agonadism, omphalocele, dextrocardia, and congenital diaphragmatic hernia; VACTERL = vertebral, anorectal, cardiac, tracheo-esophageal, renal, and limb anomalies; ADAM sequence = Amniotic Deformitis, Adhesions, Mutilations. Sometimes there are overlapping features between the presented syndromes, but the severity of the associated abnormalities, especially the cardiac ones, as well as the degree of pulmonary hypoplasia are far more indicative of prognosis than any specific classification system. In counseling families to decide on a termination of pregnancy or when planning for delivery and early management, a descriptive antenatal diagnosis is more important than trying to line characteristics up with a specific syndrome.

## Data Availability

Not applicable.

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
