# Peer review of "Omphalocele and Cardiac Abnormalities—The Importance of the Association"

_diagnostics, 2023, doi:10.3390/diagnostics13081413_

Round 1

Reviewer 1 Report

The relevance and methodology of the study are questionable. I would advice the authors to follow PRISMA guidelines and fill in the PRISMA flow chart for their review. If the aim of the study is 'to highlight, through a review of the literature, the importance and 68 frequency of association between the two malformations and what impact this associa- 69 tion has on the management and evolution of patients with these pathologies' the available literature should be searched systematically.

Author Response

Dear Reviewer,

Thank you very much for evaluating our manuscript. Your recommendations and comments have helped us greatly improve our manuscript.

We followed PRISMA guidelines and fill in the PRISMA flow chart for our review (Figure 1). We systematically searched the available literature from three medical databases, adding more information and 15 new references. We added a table listing the defects and percentages/mortality associated with the cardiac defect and also a Limitation of the study paragraph.

Here we provide the requested corrections and address the comments. The changes we have made in the manuscript are highlighted in red.

Thank you again for reviewing our manuscript,

Elena Țarcă, MD, PhD

Reviewer 2 Report

Thank you for allowing me to review this paper on Omphalocele and congenital heart defects.  I do believe this is important information for physicians in these fields to know about. I do believe a few things need to be adjusted to optimize the presentation.

1. In the abstract the first sentence needs to be changed into two or more sentences as it is very wordy.

2. In the second sentence you comment about the aim of this study however this would best be worded as paper since this is a review.

3. The second sentence in the introduction section needs to be reworded as it does not fully make sense. I would reword it as "The association of several birth defects can be even more serious as it potentially increases morbidity, infant mortality as well as increasing medical costs and prolonging hospitalization."

4. On line 57 Anyway is unnecessary and should be deleted.

5. line 115 states may be diagnosticated. This should be deleted as it is redundant.

6. The sentence starting on line 137 is a run-on sentence and needs to be broken into more sentences to make a more valid point.

7. line 144 start with Is relevant in this way. This phrase is not needed.

8. Line162 death rate should be ,mortality.

9.  The paragraph on line 180 is not needed and should be deleted.

10. Line 186 This sentence does not make sense and needs to be reworded.

11. Line 191 the sentence starting with But should be "Large VSD's usually become manifest by 3-4 weeks of life causing  congestive heart failure.

12. Line 398, the first part of the sentence is pertinent but after the comma the remainder of the sentence is not necessary.

13. Line 401 that sentence should be broken into multiple sentences.

14. Line 408 needs to be reworded as it does not make sense.

15.  Line 470 "it is an unwritten rule" this is conjecture and should be deleted.

16. Line 487 Need to give a mortality rate for omphalocele associated with other defects.

17. Line 494 should have a mortality rate given.

18. There are multiple types of grammatical errors that should be corrected.

19. I feel that this article would benefit from a table listing the defects and percentages/mortality associated with the cardiac defect.

Author Response

Dear Reviewer,

Thank you very much for evaluating our manuscript. Your recommendations and comments have helped us greatly improve our manuscript. Here we provide the requested corrections and address the comments. The changes we have made in the manuscript are highlighted in red.

  1. In the abstract the first sentence needs to be changed into two or more sentences as it is very wordy.

Response: We modified the first sentence in the abstract.

  1. In the second sentence you comment about the aim of this study however this would best be worded as paper since this is a review.

Response: We modified the sentence in the abstract and in the Introduction.

  1. The second sentence in the introduction section needs to be reworded as it does not fully make sense. I would reword it as "The association of several birth defects can be even more serious as it potentially increases morbidity, infant mortality as well as increasing medical costs and prolonging hospitalization."

Response: We reworded the phrase as you suggested. Thank you.

  1. On line 57 Anyway is unnecessary and should be deleted.

Response: We deleted the unnecessary word.

  1. line 115 states may be diagnosticated. This should be deleted as it is redundant.

Response: We corrected.

  1. The sentence starting on line 137 is a run-on sentence and needs to be broken into more sentences to make a more valid point.

Response: We corrected.

  1. line 144 start with Is relevant in this way. This phrase is not needed.

Response: We corrected.

  1. Line162 death rate should be ,mortality.

Response: We corrected.

  1. The paragraph on line 180 is not needed and should be deleted.

Response: We deleted the paragraph.

  1. Line 186 This sentence does not make sense and needs to be reworded.

Response: We reworded the sentence.

  1. Line 191 the sentence starting with But should be "Large VSD's usually become manifest by 3-4 weeks of life causing congestive heart failure.

Response: We reworded the sentence as you suggested. Thank you.

  1. Line 398, the first part of the sentence is pertinent but after the comma the remainder of the sentence is not necessary.

Response: We reformulated the sentence.

  1. Line 401 that sentence should be broken into multiple sentences.

Response: We corrected.

  1. Line 408 needs to be reworded as it does not make sense.

Response: We reformulated the sentence.

  1. Line 470 "it is an unwritten rule" this is conjecture and should be deleted.

Response: We deleted the unnecessary words.

  1. Line 487 Need to give a mortality rate for omphalocele associated with other defects.

Response: We added the mortality rate and the appropriate reference.

  1. Line 494 should have a mortality rate given.

Response: We added the mortality rate and the appropriate reference.

  1. There are multiple types of grammatical errors that should be corrected.

Response: We revised the manuscript and corrected the errors.

  1. I feel that this article would benefit from a table listing the defects and percentages/mortality associated with the cardiac defect.

Response: We added the required table. Thank you for your suggestion.

Thank you again for reviewing our manuscript,

Elena Țarcă, MD, PhD

Round 2

Reviewer 1 Report

Thank you for the revision.